# Non-linear association of liver enzymes with cognitive performance in the elderly: A cross-sectional study

Yan-Li Zhang[1], Shi-Ying Jia[2], Bo Yang[3], Jie Miao[2], Chen Su[2], Zhi-Gang Cui[4], Li-Ming Yang[1]*, Jun-Hong Guo[2]*

1 Department of Neurological Intensive Care Unit, Sixth Hospital of Shanxi Medical University (General Hospital of Tisco), Taiyuan, Shanxi, China, 2 Department of Neurology, First Hospital of Shanxi Medical University, Taiyuan, Shanxi, China, 3 Department of Hernia and Abdominal Wall Surgery, Shanxi Bethune Hospital, Shanxi Academy of Medical Sciences, Tongji Shanxi Hospital, Third Hospital of Shanxi Medical University, Taiyuan, Shanxi, China, 4 Department of Neurology, The Third People's Hospital of Datong, Datong, Shanxi, China

* neuroguo@163.com (J-HG); tgylm1970@163.com (L-MY)

**Data Availability Statement:** The datasets utilized in the present study can be accessed from the NHANES and are available at https://wwwn.cdc.gov/nchs/nhanes/.

## Abstract

### Background

Although liver metabolic dysfunction has been found to potentially elevate susceptibility to cognitive impairment and dementia, there is still insufficient evidence to explore the non-linear association of liver enzymes with cognitive performance. Therefore, we aimed to elucidate the non-linear relationship between liver enzymes and cognitive performance.

### Methods

In this cross-sectional study, 2764 individuals aged $\geq$ 60 who participated in the National Health and Nutrition Survey (NHANES) between 2011 and 2014 were included. The primary data comprised liver enzyme levels (alkaline phosphatase (ALP), aspartate aminotransferase (AST), alanine aminotransferase (ALT), AST/ALT ratio, and gamma-glutamyl transferase (GGT)), and cognitive performance was the major measured outcome. The associations were analyzed using weighted multivariate logistic regression, subgroup analysis, a generalized additive model, smooth fitting curves, and threshold effects.

### Results

The results of the fully adjusted model indicated that ALP was negatively associated with the animal fluency test (AFT) score (OR = 1.48, 95% CI: 1.11–1.98), whereas ALT demonstrated a positive association with the consortium to establish a registry for Alzheimer's disease (CERAD) test score (OR = 0.72, 95% CI: 0.53–0.97). Additionally, the AST/ALT ratio was negatively associated with the global cognitive test (OR = 2.39, 95% CI: 1.53–3.73), CERAD (OR = 2.61, 95% CI: 1.77–3.84), and digit symbol substitution test (DSST) scores (OR = 2.51, 95% CI: 1.57–4.02). GGT was also negatively associated with the AFT score (OR = 1.16, 95% CI: 1.01–1.33) in unadjusted model. A non-linear relationship was observed between liver enzymes and the risk of cognitive impairment as assessed by the

**Funding:** This work was supported by the "Shanxi Province Basic Research Program (202303021212373)." Yan-Li Zhang, the host of the fund project and the first author of this article, played a crucial role in study design, data collection and analysis, decision to publish, and preparation of the manuscript.

**Competing interests:** The authors have declared that no competing interests exist.

**Abbreviations:** AD, Alzheimer's disease; AFT, the animal fluency test; ALB, albumin; ALP, alkaline phosphatase; ALT, alanine aminotransferase; AST, aspartate aminotransferase; BMI, body mass index; CERAD, : consortium to establish a registry for Alzheimer's disease; CHD, coronary heart disease; CI, confidence interval; DSST, digit symbol substitution test; GAM, generalized additive model; GGT, gamma-glutamyl transpeptidase; NHANES, National Health and Nutrition Survey; ORs, Odds ratios; PIR, : poverty–income ratio; SD, standard deviation; SCr, : serum creatinine; SUA, serum uric acid; TB, total bilirubin; TC, total cholesterol; TG, triglyceride.

global cognitive test. Specifically, when ALP > 60 U/L, 0.77 < AST/ALT < 1.76, and 25 < GGT < 94 U/L, higher liver enzyme levels were significantly associated with an elevated cognitive impairment risk, while a lower cognitive impairment risk when ALT level was > 17 U/L.

## Conclusions

There is a non-linear relationship between liver enzymes and cognitive performance, indicating that liver enzyme levels should be maintained within a certain level to mitigate the risk of cognitive impairment.

## 1 Introduction

The decline in cognitive performance among older adults has emerged as a significant global public health concern due to an increased life expectancy and the rise in chronic comorbidities [1]. According to the World Alzheimer Report of 2022, the global prevalence of individuals living with dementia is projected to rise to 139 million by 2050 (available online: https://www.who.int/news-room/fact-sheets/detail/dementia). Cognitive impairment manifests as diverse symptoms, like memory disorders, language deterioration, and impaired executive function, and is regarded as a substantial predictor for dementia development [2,3]. Consequently, it is imperative to identify modifiable factors that potentially contribute to cognitive decline to facilitate the development of effective preventive interventions [4–6].

Alzheimer's disease (AD) is the most common form of senile dementia [7]. Multiple studies have demonstrated that metabolic dysfunctions, including disturbances in energy metabolism, chronic inflammation, oxidative stress, and neuronal insulin resistance, play a crucial role in the pathogenesis of AD [7–10]. AD is a systemic metabolic disorder, and metabolic processes in the peripheral organs are also significant [9,10]. Hepatic metabolic activity determines the metabolic readout in the peripheral circulation [11], as reported in a previous study, where hepatic dysfunction is proposed to contribute to AD due to failure to maintain Aβ homeostasis in the periphery, consequently resulting in the release of proinflammatory cytokines due to chronic inflammation or metabolic dysfunction [12]. Therefore, impairments in hepatic metabolic activities are associated with the development of cognitive impairment and dementia [7,12,13].

Evaluation of hepatic function entails quantifying the levels of alkaline phosphatase (ALP), alanine aminotransferase (ALT), aspartate aminotransferase (AST), AST/ALT ratio, and gamma-glutamyl transferase (GGT) in the peripheral blood. A significant correlation has been reported between elevated ALP levels and impaired cognitive function [7,14]. Besides, ALT and AST are associated with cardiovascular and metabolic disorders [14,15] which are recognized risk factors for AD and cognitive decline [16,17]. Similarly, lower ALT levels and elevated AST/ALT ratios were reported to be associated with diminished cerebral glucose metabolism, impaired neurotransmitter production and synaptic transmission, and elevated amyloid-β deposition [7,8]. GGT has been recognized as a biomarker for oxidative stress and an atherosclerosis prognosticator [18,19]. The initial GGT level and GGT variability were positively and independently linked to the prospective risk of dementia [20–22].

However, the extent to which cognitive impairment is linked to altered liver enzymes remains inadequately investigated, and a definitive threshold for the impact of each liver enzyme on cognitive function has yet to be established. Consequently, we conducted a cross-

sectional study utilizing the National Health and Nutrition Examination Survey (NHANES) 2011–2014 database to investigate the association between various liver enzymes and cognitive function and evaluate the potential non-linear associations between different liver enzymes and cognitive function in geriatrics.

## 2 Methods

### 2.1 Study population

The data used in this study were obtained from the 2011–2014 cycles of the NHANES, a comprehensive nationwide cross-sectional survey conducted by the Centers for Disease Control and Prevention (CDC). Comprehensive information regarding the survey contents and sampling methods is given elsewhere [23,24]. The NHANES cycles were approved by the Research Ethics Review Board (ERB) of the National Center for Health Statistics (NCHS) at the CDC. Written consent was obtained from all participants in the survey [25].

Among the 19,931 individuals in the NHANES 2011–2014 database, a subset of 3632 individuals were identified as being ≥ 60 years old and thus potentially suitable for cognitive functioning evaluation. After excluding 868 individuals due to incomplete cognitive function tests or liver enzyme measurements, 2764 participants were included in this study, representing a weighted population estimate of approximately 50.1 million non-institutionalized U.S. adults aged ≥ 60 (Fig 1).

### 2.2 Liver Enzymes measurement

Beckman Coulter UniCel DxC800 (Beckman Coulter) was employed for detecting and measuring levels of liver enzymes, including albumin, ALP, AST, ALT, GGT, and total bilirubin (TB) [26]. ALP and ALT activities were measured using a kinetic rate method, while AST and GGT were determined using an enzymatic rate method. Albumin concentration was measured using the bichromatic digital endpoint method, while TB concentration was determined using a timed-endpoint diazo method known as Jendrassik-Grof. More information is available at https://wwwn.cdc.gov/Nchs/Nhanes/2011-2012/BIOPRO_G.htm and https://wwwn.cdc.gov/Nchs/Nhanes/2013-2014/BIOPRO_G.htm.

### 2.3 Assessment of cognitive performance

The cognitive function of participants aged ≥ 60 was assessed using three tests (available online: https://wwwn.cdc.gov/Nchs/Nhanes/2011-2012/CFQ_G.htm and https://wwwn.cdc.gov/Nchs/Nhanes/2013-2014/CFQ_G.htm), i.e., word learning and recall modules from the consortium to establish a registry for Alzheimer's disease (CERAD) test, the animal fluency test (AFT), and the digit symbol substitution test (DSST).

CERAD test evaluates the capacity to acquire and retain new verbal information [27,28], comprising three consecutive learning trials (CERAD-IR, consortium to establish a registry for Alzheimer's Disease immediate recall) and a delayed recall task (CERAD-DR, consortium to establish a registry for Alzheimer's Disease delayed recall). During the learning trials (CERAD-IR), participants were prompted to recall as many words as possible immediately after being instructed to articulate 10 unrelated words. The delayed word-recall task (CERAD-DR) occurred after the completion of the AFT and DSST. The overall score was determined by summing the results of CERAD-IR and CERAD-DR. The AFT test assesses categorical verbal fluency, a constituent of executive function [27,28]. Participants were instructed to name as many animals as possible within one minute, and the final score was determined based on the cumulative number of accurate answers. The DSST evaluates cognitive abilities such as

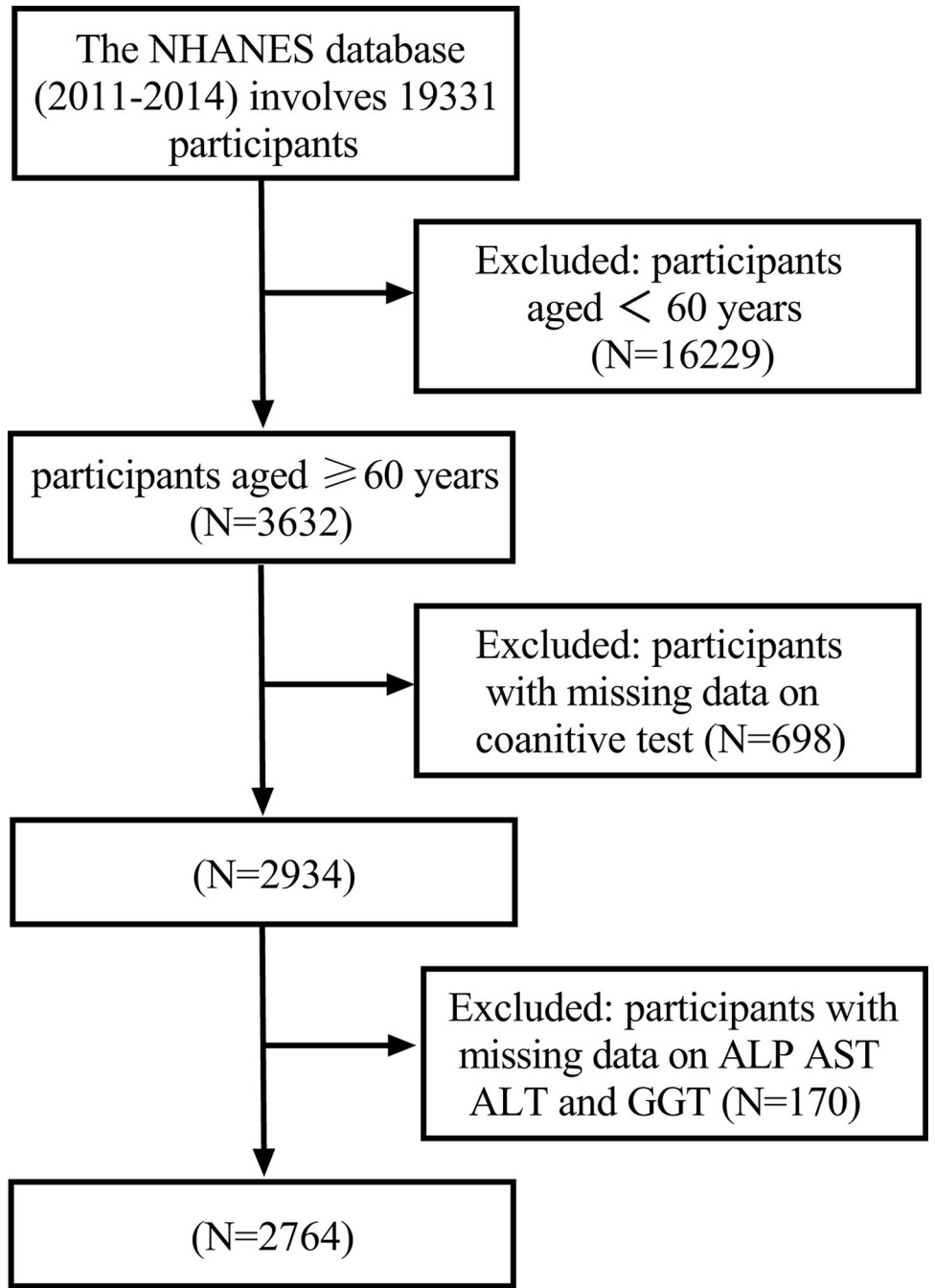

**Fig 1. Flow chart for screening study participants from NHANES 2011–2014.**

processing speed, sustained attention, and working memory [29,30]. Each participant strived to replicate the corresponding symbols accurately in 133 boxes adjacent to the numbers within 2 min, and the DSST score referred to the total number of correct matches, where a higher score on each cognitive test indicated superior cognitive performance.

Considering the lack of an acknowledged standard for determining low cognitive performance in the CERAD, AFT, and DSST assessments, the lowest quartile as a reference point was selected to use for discerning different cognitive impairment types, which was consistent

with the approaches adopted in previous studies [27,29,30]. Moreover, given the observed significance of age's impact on cognitive performance, the three cognitive test scores were subsequently adjusted based on age categories ($\geq$ 60 and $\geq$ 70 years) [27,28,30], as indicated in S1 Table. Participants whose scores were at or below the cut-off points were categorized as having poor cognitive performance, whereas those with scores above the cut-off points were classified as having normal cognitive performance. Finally, considering the potential floor and ceiling effects in individual cognitive scores [31,32], a global cognitive performance score was derived by averaging the standardized z-scores obtained from the three cognitive tests. The z-score was calculated using the formula z = (x-m)/σ, where x represents the raw score, m denotes the overall mean, and σ is the standard deviation (SD) [27,28,32].

### 2.4 Covariates

Our study incorporated previously reported variables affecting liver and cognitive function and other variables accumulated through clinical experience [27,28,33]. Following the guidelines provided by the STROBE statement, introducing covariates results in a modification of the basic model by more than 10% [34]. Hence, we included the subsequent covariables, i.e., gender, race (Mexican American, other Hispanic, non-Hispanic White, non-Hispanic Black, or other race), age ($\geq$ 60 and $\geq$ 70 years), education level (below high school, high school, and above high school), poverty–income ratio (PIR) ($<$ 1.3 and $\geq$ 1.3), physical activity (vigorous ($\geq$ the recommended level of activity), moderate ($<$ the recommended level of activity) and no activity), as supported by evidence suggesting that Americans advised to engage in at least 75 min of vigorous or 150 min of moderate physical activity/week [35], smoking (current smokers (smoke $\geq$ 100 cigarettes and continue), never smokers ($<$ 100 cigarettes in their lifetime) and former smokers (smoked $\geq$ 100 cigarettes in their lifetime but currently no smoking)), body mass index (BMI) ($<$ 25 kg/m$^2$; 25 to 30 kg/m$^2$; $\geq$ 30 kg/m$^2$), alcohol consumption (none, moderate (defined as $>$ 0 to $\leq$ 2 drinks/day for men or $>$ 0 to $\leq$ 1 drink/day for women), or heavy (defined as $>$ 2 to $<$ 5 drinks/day for men or $>$ 1 to $<$ 4 drinks/day for women)) [36], and chronic disease conditions including diabetes (defined by questionnaire "Doctor told you have diabetes" and/or laboratory measured glycated hemoglobin A1c (HbA1c) $\geq$ 6.5%), hypertension (defined by questionnaire "Have you ever been diagnosed with hypertension by a healthcare professional?" Have you been advised to take prescribed medication for high blood pressure?" "Are you currently taking prescribed medication for hypertension?" and systolic blood pressure $\geq$ 140 and/or diastolic blood pressure $\geq$ 90 mmHg), stroke (defined by a questionnaire "Have you ever received a medical diagnosis of stroke?"); coronary heart disease (CHD, determined by a questionnaire "Has a medical professional ever told you that you had a CHD?"), and liver disease (determined by the questionnaire "Have you ever been told by a doctor about the presence of any liver ailment?"). Total cholesterol (TC, mmol/L), triglyceride (TG, mmol/L), and serum uric acid (SUA, μmol/L) levels were measured in the laboratory.

### 2.5 Statistical analysis

Statistical analysis was conducted on the original datasets obtained from NHANES. Following the NHANES Weight Analytical Guide [37], the MEC exam sample weights (WTMEC2YR) were employed to calculate the new sample weights after merging the datasets from 2011 to 2014. After log2 transformation, liver enzymes exhibited a normal distribution [7]. Continuous variables were assessed by calculating the weighted mean±SD, whereas comparisons between groups with low and normal cognitive performance were performed using a weighted

linear regression model. Categorical variables were presented as n (%) and were compared using the weighted Chi-square test.

According to the STROBE statement [34], a multivariate test was conducted using three models, i.e., Model 1: no covariates adjusted, Model 2: adjusted for age, gender, race, education status, and PIR, and Model 3: adjusted for all covariates. Weighted multivariate logistic regression models were developed to examine the association between liver enzyme levels and cognitive performance. Subsequently, liver enzyme levels were categorized into four groups based on quartiles, ranging from the lowest (Q1) to the highest (Q4). Odds ratios (ORs) and 95% confidence intervals (CIs) were calculated for all three models. Moreover, weighted multivariate logistic regression was employed to conduct a subgroup analysis. If the interaction *P*-value yielded insignificance, the outcomes of the distinct strata could be deemed reliable. However, if significance was observed, it indicated the presence of a distinct population subset. In addition, we employed a generalized additive model (GAM) based on fitted smooth curves and a threshold effects analysis model to examine the non-linear relationships and identify inflection points between liver enzymes and the risk of cognitive impairment. All statistical analyses were performed using EmpowerStats (version 2.0, www.empowerstats.com) and Stata v17.0, with $P < 0.05$ indicated statistically significant.

## 3 Results

### 3.1 Demographic characteristics of study participants

The demographic characteristics of the included participants in the liver enzyme survey concerning different dimensions of cognitive performance are presented in Tables 1 and S2. A total of 2764 individuals aged $\geq 60$ participated in this study, with 2045 exhibiting normal cognitive function and 719 demonstrating decreased cognitive function, assessed by global cognitive performance. Results indicated a significant correlation between cognitive function and various factors, including age, race, educational status, income level, BMI (except for CREAD), physical activity, alcohol consumption, presence of chronic diseases (including diabetes, hypertension, and stroke), ALB, ALP, ALT, AST/ALT ratio, GGT (in globe cognitive performance and AFT), TC and serum creatinine (SCr) levels ($P < 0.05$). The underlying reason for cognitive impairment lies in various sociological, disease-related, and lifestyle factors. Among the sociological factors, advanced age, male, and low educational level play a significant role. Disease-related factors, including diabetes, hypertension, and coronary heart disease, also contribute to this condition. Furthermore, unhealthy lifestyle habits such as excessive drinking and inadequate physical activity are closely associated with the development of cognitive impairment. In our study, the proportion of cognitive impairment is higher in participants who were male, older ($\geq 70$ years), Mexican American, other Hispanic, non-Hispanic Black; have lower educational level, lower poverty–income ratio, and lower BMI; not engaged in physical activity; current smokers and higher alcohol consumers. Additionally, this group displayed a higher prevalence of chronic conditions such as diabetes, hypertension, and stroke. In terms of biochemical markers, individuals with cognitive impairment demonstrated reduced concentrations of ALB, ALT, and SCr, alongside elevated levels of ALP, AST/ALT ratio, GGT, and TC. However, no noteworthy disparities were noted in AST levels between the normal and impaired cognitive groups. For CREAD and DSST, participants who were male, with coronary heart and liver disease had lower cognitive performance.

### 3.2 Association between liver enzymes and cognitive performance

**3.2.1 Relationship between ALP and cognitive performance.** The association results between ALP and various dimensions of cognitive performance in geriatric participants are

**Table 1. Characteristics of liver enzymes and the globle cognitive performance (assessed by z-score) among participants from NHANES 2011–2014 (N = 2764).**

| Variable | Total (N = 2764) | Global Cognitive Performance (Zscore) | | P value |
| --- | --- | --- | --- | --- |
| | | Normal cognitive performance (N = 2045) | Low cognitive performance (N = 719) | |
| Gender | | | | 0.115 |
| Male | 1357 (49.1%) | 954 (46.7%) | 403 (56.1%) | |
| Female | 1407 (50.9%) | 1091 (53.3%) | 316 (43.9%) | |
| Age(years) | 69.2±6.6 | 68.8±6.5 | 71.1±7.1 | <0.001 |
| ≥60 | 1498 (54.2%) | 1102 (53.9%) | 396 (55.1%) | |
| ≥70 | 1266 (45.8%) | 943 (46.1%) | 323 (44.9%) | |
| RACE | | | | <0.001 |
| Mexican American | 244 (8.8%) | 152 (7.4%) | 92 (12.8%) | |
| Other Hispanic | 277 (10.0%) | 152 (7.4%) | 125 (17.4%) | |
| Non-Hispanic White | 1351 (48.9%) | 1134 (55.5%) | 217 (30.2%) | |
| Non-Hispanic Black | 627 (22.7%) | 395 (19.3%) | 232 (32.3%) | |
| Other Race | 265 (9.6%) | 212 (10.4%) | 53 (7.4%) | |
| Education status | | | | <0.001 |
| Below high school | 689 (24.9%) | 327 (16.0%) | 362 (50.3%) | |
| High school | 657 (23.8%) | 483 (23.6%) | 174 (24.2%) | |
| Above high school | 1416 (51.2%) | 1235 (60.4%) | 181 (25.2%) | |
| Not recorded | 2 (0.1%) | 0 (0.0%) | 2 (0.3%) | |
| Poverty income ratio | | | | <0.001 |
| <1.3 | 747 (27.0%) | 441 (21.6%) | 306 (42.6%) | |
| ≥1.3 | 1788 (64.7%) | 1446 (70.7%) | 342 (47.6%) | |
| Not recorded | 229 (8.3%) | 158 (7.7%) | 71 (9.9%) | |
| Body mass index(kg/m$^2$) | | | | 0.015 |
| <25 | 734 (26.6%) | 544 (26.6%) | 190 (26.4%) | |
| 25–30 | 965 (34.9%) | 708 (34.6%) | 257 (35.7%) | |
| ≥30 | 1025 (37.1%) | 772 (37.8%) | 253 (35.2%) | |
| Not recorded | 40 (1.4%) | 21 (1.0%) | 19 (2.6%) | |
| Physical activity level | | | | <0.001 |
| No | 1169 (42.3%) | 770 (37.7%) | 399 (55.5%) | |
| Moderate | 472 (17.1%) | 371 (18.1%) | 101 (14.0%) | |
| Vigorous | 1119 (40.5%) | 902 (44.1%) | 217 (30.2%) | |
| Not recorded | 4 (0.1%) | 2 (0.1%) | 2 (0.3%) | |
| Smoking staus | | | | 0.287 |
| Non-smoker | 1362 (49.3%) | 1022 (50.0%) | 340 (47.3%) | |
| Former smoker | 1043 (37.7%) | 785 (38.4%) | 258 (35.9%) | |
| Current smoker | 357 (12.9%) | 236 (11.5%) | 121 (16.8%) | |
| Not recorded | 2 (0.1%) | 2 (0.1%) | 0 (0.0%) | |
| Drinking status | | | | <0.001 |
| No | 664 (24.0%) | 457 (22.3%) | 207 (28.8%) | |
| Moderate | 1285 (46.5%) | 1023 (50.0%) | 262 (36.4%) | |
| Heavy | 768 (27.8%) | 540 (26.4%) | 228 (31.7%) | |
| Not recorded | 47 (1.7%) | 25 (1.2%) | 22 (3.1%) | |
| Diabetes(Yes) | 761 (27.5%) | 497 (24.3%) | 264 (36.7%) | <0.001 |
| Hypertention(Yes) | 917 (33.2%) | 624 (30.5%) | 293 (40.8%) | <0.001 |
| Coronary heart disease(Yes) | 256 (9.3%) | 183 (8.9%) | 73 (10.2%) | 0.264 |
| Stroke(Yes) | 192 (6.9%) | 108 (5.3%) | 84 (11.7%) | <0.001 |
| Liver disease(Yes) | 156 (5.6%) | 114 (5.6%) | 42 (5.8%) | 0.115 |

(*Continued*)

**Table 1.** (Continued)

| Variable | Total (N = 2764) | Global Cognitive Performance (Zscore) | | |
| --- | --- | --- | --- | --- |
| | | Normal cognitive performance (N = 2045) | Low cognitive performance (N = 719) | P value |
| laboratory data | | | | |
| LogALP(U/L) | 6.0±0.4 | 6.0±0.4 | 6.1±0.5 | <0.001 |
| LogAST(U/L) | 4.6±0.4 | 4.6±0.4 | 4.6±0.5 | 0.115 |
| LogALT(U/L) | 4.4±0.5 | 4.4±0.5 | 4.3±0.7 | 0.002 |
| AST/ALT | 1.2±0.3 | 1.2±0.3 | 1.3±0.4 | <0.001 |
| LogGGT(U/L) | 4.3±0.8 | 4.3±0.8 | 4.4±1.0 | 0.02 |
| ALB(g/L) | 42.1±2.9 | 42.2±2.8 | 41.4±3.4 | <0.001 |
| TB(umol/L) | 11.8±4.7 | 11.7±4.2 | 12.1±6.7 | 0.147 |
| TC (mmol/L) | 5.0±1.1 | 5.0±1.1 | 4.8±1.1 | 0.001 |
| TG(mmol/L) | 1.8±1.1 | 1.7±1.1 | 1.8±1.2 | 0.773 |
| LogSCr(umol/L) | 6.4±0.4 | 6.3±0.4 | 6.5±0.6 | <0.001 |
| SUA(umol/L) | 333.9±84.9 | 333.9±85.3 | 334.0±82.6 | 0.973 |

Continuous variables were expressed as weighted mean ± SD and analyzed using a weighted linear regression model to determine *P* values. Categorical variables were presented as n (%) and analyzed using a weighted chi-square test to assess significance. NHANES, National Health and Nutrition Examination Survey; ALP: Alkaline phosphatase; AST: Aspartate aminotransferase; ALT: Alanine aminotransferase; GGT: Gamma-glutamyl transpeptidase; ALB: Albumin; TB: Total bilirubin; TC: Total cholesterol; TG: Triglyceride; SCr: Serum creatinine; SUA: Serum uric acid.

shown in Tables 2, 3 and S3. After adjusting for all covariates, ALP levels between 55-65U/L were associated with a lower risk of cognitive impairment assessed by global cognitive performance with OR (95%CI) being 0.66 (0.44–0.99) (Table 2) and ALP levels between 66-79U/L were associated with a lower risk of cognitive impairment, as assessed by DSST with OR (95% CI) being 0.64 (0.42–0.97) (Table 3). While the fully adjusted model revealed a noteworthy negative correlation between ALP and AFT, with an OR (95% CI) of 1.48 (1.11–1.98) (Table 3). Furthermore, when treated as a categorical variable based on quartiles, ALP levels higher than 80U/L were associated with an elevated risk of cognitive impairment assessed by AFT with OR (95%CI) being 1.50 (1.06–2.12) after adjusting for age, gender, race, education status, and PIR (S3 Table). These contradictory findings suggest a non-linear association between ALP and cognitive function, indicating an inflection point in their relationship.

In the sub-groups analysis (S4 Table) stratified by gender, age, race, education, physical activity, alcohol consumption, smoking, and chronic diseases (hypertension, diabetes, stroke, and CHD), the findings revealed a consistent association between ALP and global cognitive performance in individuals who were identified as Mexican American and those with a low level of education (*P* for trend < 0.01). The influence of race on the relationship between ALP and global cognitive performance was found to be significant (*P* for interaction < 0.001). Specifically, Mexican Americans demonstrated a strong correlation between ALP and the likelihood of cognitive impairment compared to those of other racial backgrounds.

**3.2.2 Relationship between ALT and cognitive performance.** Tables 2, 3 and S3 present the association outcomes between ALT and diverse aspects of cognitive performance among geriatric individuals. In the fully adjusted model, a significant positive association was observed between ALT and CERAD test, yielding an OR (95% CI) of 0.72 (0.53–0.97) (Table 3). Moreover, the statistical significance of this trend remained evident even when ALT was analyzed as a categorical variable (quartile). Adjusting for all covariates, the highest ALT quartile consistently demonstrated a lower likelihood of impaired cognitive function compared to the lowest quartile, with OR (95%CI) values of 0.55 (0.37–0.82) for global cognitive

**Table 2. The associations between liver enzymes and Global Cognitive Performance among participants from NHANES 2011–2014 (N = 2764).**

| Variable(U/L) | Globe Cognitive Performance | | |
|---|---|---|---|
| | Model 1 OR(95%CI) | Model 2 OR(95%CI) | Model 3 OR(95%CI) |
| LogALP | 1.79***(1.31–2.43) | 1.39(0.98–1.99) | 1.27(0.89–1.79) |
| ALP(Quartile) | | | |
| Q1(14–54) | 1.00(Ref.) | 1.00(Ref.) | 1.00(Ref.) |
| Q2(55–65) | 0.83(0.58–1.18) | 0.70(0.48–1.04) | 0.66*(0.44–0.99) |
| Q3(66–79) | 1.22(0.86–1.72) | 1.00(0.68–1.47) | 1.03(0.68–1.55) |
| Q4(80–336) | 1.82***(1.30–2.54) | 1.35(0.92–1.98) | 1.22(0.83–1.81) |
| *P* for trend | <0.001 | 0.055 | 0.119 |
| LogALT | 0.72(0.51–1.01) | 0.79(0.55–1.12) | 0.82(0.58–1.15) |
| ALT(Quartile) | | | |
| Q1(5–15) | 1.00(Ref.) | 1.00(Ref.) | 1.00(Ref.) |
| Q2(16–19) | 0.42***(0.30–0.57) | 0.44***(0.30–0.62) | 0.46***(0.32–0.66) |
| Q3(20–24) | 0.33***(0.24–0.45) | 0.36***(0.25–0.52) | 0.39***(0.27–0.57) |
| Q4(25–228) | 0.49***(0.35–0.69) | 0.52**(0.35–0.77) | 0.55**(0.37–0.82) |
| *P* for trend | <0.001 | 0.002 | 0.005 |
| LogAST/ALT | 2.46***(1.71–3.54) | 2.59***(1.71–3.91) | 2.39***(1.53–3.73) |
| AST/ALT(Quartile) | | | |
| Q1(0.26–0.99) | 1.00(Ref.) | 1.00(Ref.) | 1.00(Ref.) |
| Q2(1.00–1.17) | 1.06(0.72–1.55) | 1.08(0.70–1.65) | 1.03(0.66–1.60) |
| Q3(1.18–1.37) | 1.09(0.75–1.58) | 1.10(0.73–1.64) | 1.05(0.69–1.62) |
| Q4(1.38–5.12) | 2.06***(1.44–2.96) | 2.23***(1.47–3.36) | 2.00**(1.28–3.11) |
| *P* for trend | <0.001 | <0.001 | 0.002 |
| LogGGT(U/L) | 1.15(0.99–1.33) | 1.06(0.89–1.26) | 1.05(0.87–1.26) |
| GGT(Quartile) | | | |
| Q1(5–13) | 1.00(Ref.) | 1.00(Ref.) | 1.00(Ref.) |
| Q2(14–18) | 0.99(0.70–1.38) | 0.94(0.65–1.35) | 0.95(0.65–1.40) |
| Q3(19–27) | 0.87(0.62–1.24) | 0.70(0.47–1.02) | 0.72(0.48–1.07) |
| Q4(28–423) | 1.19(0.84–1.66) | 0.89(0.61–1.29) | 0.87(0.58–1.30) |
| *P* for trend | 0.504 | 0.283 | 0.267 |

Weighted binary logistic regression analyses were used to caculate weighted ORs and 95% CIs. Model 1 adjusted for no covariates. Model 2 adjusted for age, gender, race, education status, and PIR. Model 3 adjusted for gender, race, age, education level, PIR, BMI, physical activity, smoking, drinking, diabetes, hypertension, stroke, coronary heart disease, liver disease, TC, TG, and SUA. * *P* < 0.05

** *P* < 0.01

*** *P* < 0.001.

performance (Table 2), 0.53 (0.37–0.76) for CERAD (Table 3), and 0.53 (0.34–0.80) for DSST (Table 3) (*P* for trend < 0.01). These findings reveal a strong dose-dependent association between cognitive performance and ALT levels, indicating that a higher ALT dosage conferred a more pronounced cognitive protective effect.

In the subgroup analysis (S5 Table), a stable association between ALT and global cognitive performance was observed among specific demographic groups, including males, individuals ≥ 70 years, individuals of other Hispanic and non-Hispanic White ethnicities, those with lower and higher levels of education, individuals with no physical activity, moderate alcohol consumption, and with chronic diseases (*P* for trend < 0.05). However, it was found

**Table 3. The associations between liver enzymes and different dimensions of cognitive performance (CERAD Test, AFT and DSST) in Model 3 among participants from NHANES 2011–2014 (N = 2764).**

| Variable (U/L) | CERAD Test | AFT | DSST |
|---|---|---|---|
| | Model 3 | Model 3 | Model 3 |
| | OR(95%CI) | OR(95%CI) | OR(95%CI) |
| LogALP | 1.19(0.88–1.62) | 1.48**(1.11–1.98) | 0.99(0.67–1.47) |
| ALP(Quartile) | | | |
| Q1(14–54) | 1.00(Ref.) | 1.00(Ref.) | 1.00(Ref.) |
| Q2(55–65) | 1.05(0.73–1.51) | 1.04(0.72–1.49) | 0.67(0.43–1.03) |
| Q3(66–79) | 1.21(0.84–1.76) | 1.12(0.77–1.61) | 0.64*(0.42–0.97) |
| Q4(80–336) | 1.22(0.86–1.74) | 1.37(0.96–1.96) | 0.87(0.56–1.35) |
| *P* for trend | 0.195 | 0.081 | 0.587 |
| LogALT | 0.72*(0.53–0.97) | 0.95(0.72–1.26) | 0.75(0.51–1.11) |
| ALT(Quartile) | | | |
| Q1(5–15) | 1.00(Ref.) | 1.00(Ref.) | 1.00(Ref.) |
| Q2(16–19) | 0.58**(0.41–0.81) | 0.65**(0.47–0.89) | 0.48***(0.32–0.72) |
| Q3(20–24) | 0.48***(0.33–0.68) | 0.68*(0.47–0.98) | 0.38***(0.26–0.56) |
| Q4(25–228) | 0.53**(0.37–0.76) | 0.78(0.54–1.11) | 0.53**(0.34–0.80) |
| *P* for trend | 0.001 | 0.263 | 0.003 |
| LogAST/ALT | 2.61***(1.77–3.84) | 1.34(0.91–1.97) | 2.51***(1.57–4.02) |
| AST/ALT(Quartile) | | | |
| Q1(0.26–0.99) | 1.00(Ref.) | 1.00(Ref.) | 1.00(Ref.) |
| Q2(1.00–1.17) | 1.16(0.78–1.73) | 1.08(0.72–1.61) | 0.79(0.50–1.24) |
| Q3(1.18–1.37) | 1.26(0.84–1.87) | 1.08(0.72–1.63) | 1.16(0.74–1.81) |
| Q4(1.38–5.12) | 2.21***(1.49–3.29) | 1.20(0.80–1.79) | 2.16***(1.34–3.46) |
| *P* for trend | <0.001 | 0.385 | <0.001 |
| LogGGT(U/L) | 0.98(0.84–1.14) | 1.11(0.95–1.30) | 0.96(0.80–1.16) |
| GGT(Quartile) | | | |
| Q1(5–13) | 1.00(Ref.) | 1.00(Ref.) | 1.00(Ref.) |
| Q2(14–18) | 0.98(0.69–1.38) | 1.27(0.89–1.80) | 0.72(0.47–1.11) |
| Q3(19–27) | 0.79(0.56–1.13) | 1.09(0.75–1.57) | 0.56**(0.37–0.85) |
| Q4(28–423) | 0.81(0.56–1.19) | 1.37(0.94–1.99) | 0.70(0.45–1.09) |
| *P* for trend | 0.153 | 0.234 | 0.08 |

Weighted binary logistic regression analyses were used to caculate weighted ORs and 95% CIs. Model 1 adjusted for no covariates. Model 2 adjusted for age, gender, race, education status, and PIR. Model 3 adjusted for gender, race, age, education level, PIR, BMI, physical activity, smoking, drinking, diabetes, hypertension, stroke, coronary heart disease, liver disease, TC, TG, and SUA.CERAD Test: Consortium to Establish a Registry for Alzheimer's Disease test; AFT: Animal fluency test; DSST: Digit symbol substitution test. * *P* < 0.05

** *P* < 0.01

*** *P* < 0.001.

that chronic disease conditions, particularly stroke, significantly influenced (*P* for interaction < 0.05) the association between ALT and global cognitive performance.

**3.2.3 Relationship between AST/ALT ratio and cognitive performance.** Tables 2, 3 and S3 present the association between AST/ALT levels and various cognitive outcomes among geriatric subjects. In the fully adjusted model, AST/ALT levels were significantly associated with an increased risk of cognitive impairment, with an OR (95% CI) 2.39 (1.53–3.73) for global cognitive performance (Table 2), 2.61 (1.77–3.84) for CERAD (Table 3), and 2.51 (1.57–4.02) for DSST (Table 3). The statistical significance of the trend persisted when analyzing

AST/ALT levels as a categorical variable (quartiles), with the highest quartile exhibiting a progressively increased risk of cognitive impairment compared to the lowest quartile in the fully adjusted model. Specifically, the OR (95%CI) were 2.00 (1.28–3.11) for global cognitive performance (Table 2), 2.21 (1.49–3.29) for CERAD (Table 3), and 2.16 (1.34–3.46) for DSST (Table 3) (P for trend < 0.01). These findings suggested that a lower dosage of AST/ALT conferred a more significant cognitive protective advantage.

In the subgroup analysis (S6 Table), a consistent association between AST/ALT ratio and global cognitive performance was observed in specific demographic groups, including males, individuals ≥ 70 years, those of other Hispanic and non-Hispanic White ethnicity, individuals with a higher education level, moderate physical activity, moderate alcohol consumption, non-smokers and former smokers, and those with hypertension and diabetes (P for trend < 0.05).

**3.2.4 Relationship between GGT and cognitive performance.** The results in S3 Table demonstrate a noteworthy negative association between GGT and cognitive performance in the unadjusted model for AFT, with an OR (95% CI) being 1.16 (1.01–1.33). Moreover, when considered as a categorical variable (quartiles), GGT levels in the highest quartile exhibited a negative link with AFT (OR = 1.51, 95% CI: 1.08–2.12) in unadjusted model (S3 Table). Conversely, GGT levels ranging from 19-27U/L displayed a positive correlation with DSST (OR = 0.56, 95% CI: 0.37–0.85) in the fully adjusted model (Table 3). The conflicting results hint at a complex, non-linear link between GGT and cognitive function, pointing to a potential inflection point in their association.

In the analysis of subgroups (S7 Table), a stable association between GGT and global cognitive performance was observed among individuals of other Hispanic origin with a low level of education and moderate alcohol consumption (P for trend < 0.05).

## 3.3 Non-linear relationships between liver enzymes and cognitive performance

Outliers were identified and excluded for each liver enzyme marker if their values deviated by more than 4 SDs from the mean value [7]. A sensitivity analysis was conducted in order to ascertain the robustness of the findings by excluding the outlier values of liver enzymes (S8–S11 Tables). Subsequently, the GAM and smooth curve fitting technique were employed to identify the non-linear associations between various liver enzymes and the risk of cognitive impairment, as assessed by global cognitive performance.

In the fully adjusted model, examining the relationship between ALP and the risk of cognitive impairment revealed a non-linear association (Fig 2A), with an inflection point of 60 by threshold effect analysis (Table 4). Before reaching the inflection point, a significant negative correlation was observed between ALP level and the risk of cognitive impairment, with an OR (95% CI) of 0.97 (0.96–0.99). Conversely, following the inflection point, a significant positive correlation was identified with an OR (95% CI) of 1.01 (1.00–1.02). In other words, when ALP levels exceed 60 U/L, there is a significant association with an increased risk of cognitive impairment.

Similarly, a non-linear relationship was observed between ALT and the risk of cognitive impairment in the fully adjusted model (Fig 2B), further supported by the threshold effect analysis, which identified two inflection points at 17 and 40 (Table 4). A higher incidence of cognitive impairment was observed when ALT levels were below 17. Prior to this inflection point, there was a significant negative correlation between ALT levels and the risk of cognitive impairment, with an OR (95% CI) of 0.88 (0.83–0.93). However, no significant association was found between ALT levels and cognitive impairment risk within the range of 17 to 40 (OR: 1.00; 95% CI: 0.98–1.02) or beyond the inflection point of 40 (OR: 0.96; 95% CI: 0.91–1.01).

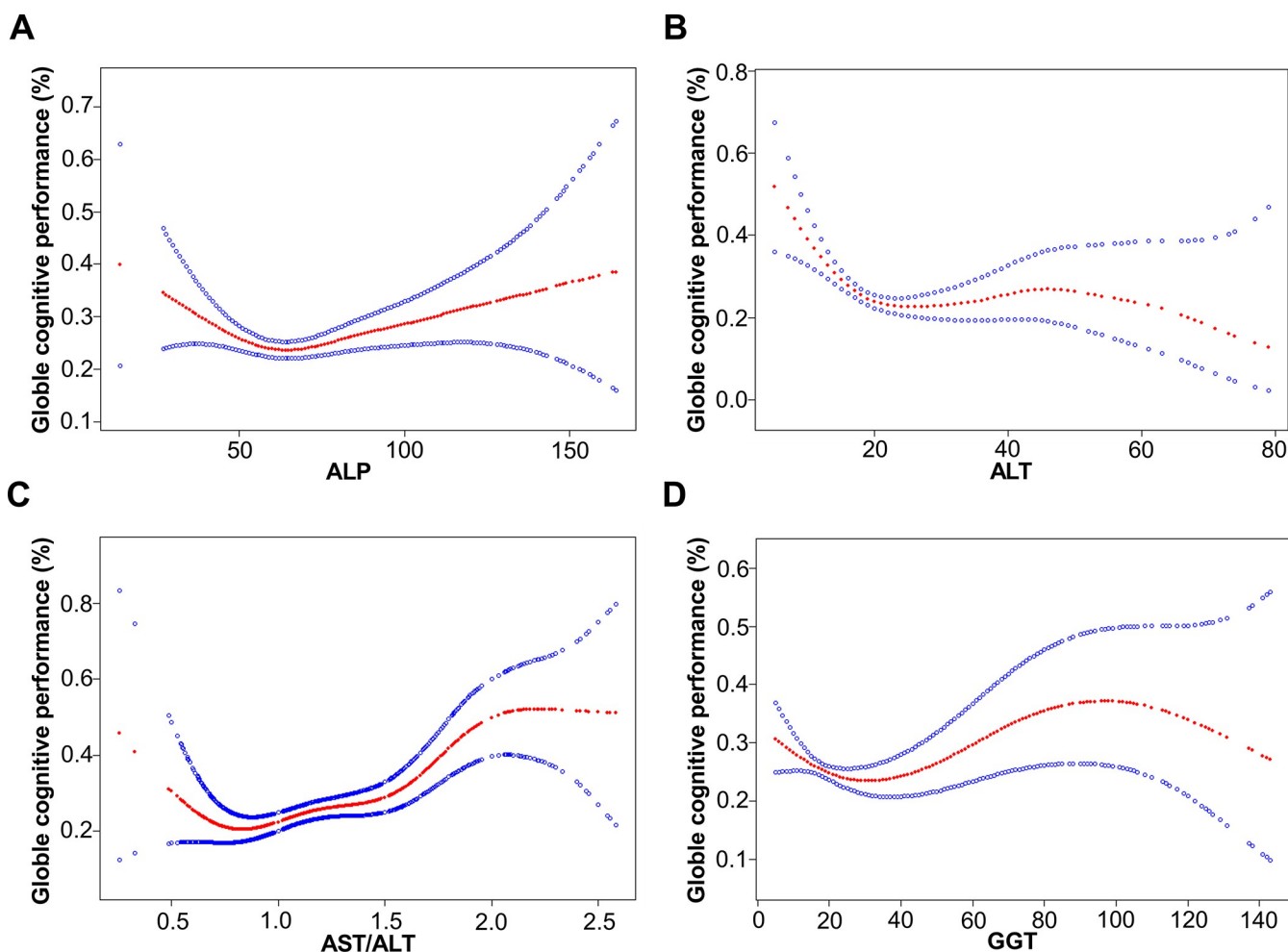

**Fig 2. Smooth curve fitting to evaluate the nonlinear relationship between liver enzymes and the risk of cognitive impairment based on the globle cognitive performance.** (A) The relationship between ALP and the risk of cognitive impairment. (B) The relationship between ALT and the risk of cognitive impairment. (C) The relationship between AST/ALT and the risk of cognitive impairment. (D) The relationship between GGT and the risk of cognitive impairment. The red solid line represents the probability of cognitive impairment occurrence and the blue dotted line represents the 95% CI curve.

**Table 4. Threshold effect analysis of liver enzymes on cognitive impairment.**

| | OR(95%CI) *P* value | | OR(95%CI) *P* value | | |
|---|---|---|---|---|---|
| | **ALP** | | **ALT** | **AST/ALT ratio** | **GGT** |
| Model I | | Model I | | | |
| Fitting by the standard liner model | 1.00 (1.00, 1.01) 0.2742 | Fiting by the standard liner model | 0.99 (0.97, 1.00) 0.0137 | 2.43 (1.76, 3.35) <0.0001 | 1.00 (1.00, 1.01) 0.2589 |
| Model II | | Model II | | | |
| Inflection point | 60 | Inflection point | 17, 40 | 0.77, 1.76 | 25, 94 |
| <inflection point | 0.97 (0.96, 0.99) 0.0027 | <inflection point1 | 0.88 (0.83, 0.93) <0.0001 | 0.01 (0.00, 0.59) 0.0249 | 0.97 (0.95, 0.99) 0.0050 |
| | | Inflection point1-2 | 1.00 (0.98, 1.02) 0.9069 | 2.27 (1.43, 3.59) 0.0005 | 1.02 (1.01, 1.03) 0.0030 |
| >inflection point | 1.01 (1.00, 1.02) 0.0026 | >inflection point2 | 0.96 (0.91, 1.01) 0.1443 | 0.99 (0.17, 5.82) 0.9897 | 0.98 (0.94, 1.03) 0.4733 |
| Log likelihood ratio | <0.001 | Log likelihood ratio | <0.001 | <0.001 | <0.001 |

That is to say, within the normal physiological range, a lower risk of cognitive impairment is associated with an ALT level exceeding 17 U/L.

Furthermore, a non-linear association was observed between the AST/ALT ratio and the risk of cognitive impairment in the fully adjusted model (Fig 2C). The threshold effect analysis revealed two inflection points at 0.77 and 1.76 (Table 4). Before the inflection point of 0.77, a significant negative correlation was found between the AST/ALT ratio and the risk of cognitive impairment, with an OR (95% CI) of 0.01 (0.00–0.59). Conversely, a significant positive correlation was observed between the inflection points of 0.77 and 1.76, with an OR (95% CI) of 2.27 (1.43–3.59). Therefore, it can be inferred that AST/ALT levels significantly correlate with increased cognitive impairment risk when the ratio falls within the range of 0.77 to 1.76.

Moreover, a non-linear association between GGT levels and the risk of cognitive impairment was observed in the fully adjusted model (Fig 2D), while the threshold effect analysis revealed two inflection points at 25 and 94 (Table 4). Before the inflection point at 25, a significant negative correlation was observed between GGT level and the risk of cognitive impairment, with an OR (95% CI) of 0.97 (0.95–0.99). Conversely, a significant positive correlation was observed between inflection points at 25 and 94, with an OR (95% CI) of 1.02 (1.01–1.03). Therefore, when GGT levels fall within the range of 25 to 94 U/L, a significant association with an increased risk of cognitive impairment can be inferred.

## 4 Discussion

Despite the longstanding association between liver enzyme status and cognitive function, a consensus regarding the specific threshold of liver enzymes for evaluating their effects on cognition is yet to be reached. In our study, following covariable screening, we utilized a multivariate bivariate logistic regression model to construct three distinct models, aiming to determine the presence of a linear relationship between cognitive function and liver enzymes. Certain findings indicated a potential correlation between elevated liver enzyme levels and a higher risk of cognitive impairment when treating liver enzymes as categorical variables. Notably, statistically significant differences were exclusively observed in the Q4 group, suggesting that the presumed linear relationship between liver enzyme levels and cognitive impairment risk may not be significant and pointing towards a potential non-linear association. Moreover, across various models, the association between varying levels of liver enzymes and cognitive impairment risk may exhibit contrasting patterns, also indicating a nonlinear relationship and an inflection point exists between liver enzyme levels and cognitive impairment risk. Furthermore, we utilize the GAM to fit a smoothing curve, allowing us to delve deeper into the nonlinear connection between liver enzyme levels and cognitive impairment risk. Additionally, we conduct an analysis of the threshold effect to identify specific threshold values for liver enzymes worthy of consideration, specifically when ALP > 60 U/L, the AST/ALT ratio fell within the range of 0.77 to 1.76, and GGT ranged from 25 to 94 U/L, a higher levels of liver enzymes were found to be significantly associated with an increased risk of cognitive impairment. Conversely, a lower incidence of cognitive impairment was observed when ALT levels > 17 U/L. These findings suggested that liver enzymes possess the potential to serve as a monitoring indicator for poor cognitive performance, emphasizing the importance of maintaining liver enzyme levels within a specific range to mitigate the risk of cognitive impairment. Further investigation is warranted in subsequent studies to reach a sound consensus regarding the suitable thresholds of liver enzyme levels concerning cognitive function.

The ALT and AST enzymes are frequently used for liver status screening [38]. Our study revealed a significant positive correlation between ALT and cognitive performance, as assessed through the global cognitive performance, CERAD test, and DSST in the fully adjusted model.

Conversely, we found a negative association between AST/ALT ratio and cognitive function, evident in the same cognitive tests after adjusting for all confounders. The findings indicated that individuals with cognitive impairment exhibited lower levels of ALT (but not AST) and a higher AST/ALT ratio than those with normal cognitive function. A subsequent stratified analysis revealed that this disparity was more pronounced among males. The present findings align with those of a previous study, wherein individuals with dementia exhibited decreased ALT levels (while AST remained unaffected) and elevated AST/ALT ratios within the normal physiological range compared to control subjects. These discrepancies were also observed when comparing men with and without dementia but were not evident among women [39], suggesting gender disparities, potentially influenced by distinct hormonal profiles [40], may contribute to liver-related metabolic processes [41].

However, contrary to prior cross-sectional studies that found strong positive links between ALT and AST levels with cognitive performance, as well as a substantial negative association between the AST/ALT ratio and cognitive function, our research revealed no statistically significant difference in AST levels between those with low and normal cognitive performance. This may be explained by the fact that AST is not exclusively produced in the liver, but also in numerous other tissues, like skeletal muscles, heart, brain. Conversely, ALT is predominantly localized in the cytoplasm of hepatocytes, making it widely regarded as the most liver-specific enzyme [8,39]. Furthermore, a previous study indicated that the AST/ALT ratio was higher in the group with cognitive impairment than in the normal cognitive function group; however, the difference in ALT levels between the two groups was insignificant [42]. Different from our findings, the ALT level of the cognitive impairment group was lower than that of the normal cognitive function group. Additionally, a recent prospective study provided evidence that low levels of plasma aminotransferases, particularly ALT, are associated with an elevated long-term risk of dementia in middle-aged patients [8].

The altered enzyme levels observed in individuals with cognitive impairments might be explained by two mechanisms. First of all, the decline in aminotransferase levels leads to a subsequent decrease in pyruvate levels, thereby reducing hepatic gluconeogenesis and glucose production [8,42], adversely affecting energy homeostasis in body tissues, including the brain [8,42]. Notably, reduced brain glucose metabolism, a distinctive feature of AD and cognitive impairment during the prodromal phase, has been documented [7]. Furthermore, it has been consistently demonstrated that elevated AST/ALT ratios and decreased ALT levels are significantly associated with cerebral glucose hypometabolism, specifically in regions that play a role in memory and executive function [7]. Second, altered ALT and AST levels may affect the production of glutamate, a major excitatory neurotransmitter involved in synaptic transmission in the cortical and hippocampal regions, consequently influencing memory and cognition through the mechanism of long-term potentiation [43,44].

The present study observed a significant negative association between serum ALP levels and cognitive performance, aligning with prior research findings [7,14,45]. However, it is pertinent to mention that our results indicated a negative association between higher ALP levels and lower executive functioning scores assessed by AFT in the fully adjusted model, partially deviating from the observed association between higher ALP levels and lower memory scores [7]. However, after adjusting for all covariates, ALP levels in the second quartile were linked to a reduced risk of cognitive impairment in global cognitive performance, while levels in the third quartile were associated with a lower risk of cognitive impairment in DSST. These conflicting results suggest a non-linear relationship between ALP and cognitive function, indicating a potential turning point in their association. Although ALP is predominantly expressed in the liver and kidneys, it is also available in the brain, specifically within endothelial cells, synaptic contacts, and the cerebral cortex [46,47]. The neuronal form of ALP plays a role in

developmental plasticity and cortical function through its involvement in γ-aminobutyric acid metabolism [48,49]. Alterations in plasma ALP levels may arise from central nervous system injury [50]. Tissue nonspecific ALP facilitates the conversion of extracellular hyperphosphory-lated monomeric tau into dephosphorylated tau, which in turn activates muscarinic receptors, leading to a sustained influx of calcium, ultimately contributing to further neuronal degeneration [51–53].

GGT level was examined and adopted to evaluate liver function [20,22]. Our findings regarding the association between GGT levels and cognitive performance were consistent with prior investigations conducted in diverse cohorts [20–22]. Futher, our findings indicate that higher GGT levels were negatively associated with executive functioning scores assessed by AFT in the unadjusted model, whereas moderate GGT levels positively correlated with DSST scores in the fully adjusted model. These conflicting observations suggest a complex, non-linear relationship between GGT and cognitive function, indicating a potential turning point in their association. GGT plays a significant role in the metabolism of glutathione and facilitates pro-oxidant and pro-inflammatory mechanisms that contribute to the development of age-related neurodegenerative disorders, including dementia [21,22,54]. Moreover, increased GGT levels have been linked to the development of atherosclerosis through their direct participation in the formation of atheromatous plaques, which have also been identified as a potential underlying factor in dementia pathogenesis [19,55]. Consequently, GGT holds promise as a potential contributor to the pathogenesis of dementia, given its role as an indicator of oxidative stress and atherosclerosis.

The primary advantage of our study lies in utilizing the NHANES database, which facilitates the acquisition of a sufficiently large sample size. Additionally, we employed three standard cognitive tests to generate a comprehensive cognitive score, thereby mitigating the influence of ceiling and floor effects and enabling the determination of cut-off values (specifically, the lowest quartile) for each age-stratified group. Furthermore, we conducted a stratified analysis to investigate the potential influence of confounding variables on the observed associations. Finally, we employed smoothed fitting curves and conducted a threshold effects analysis to investigate non-linear associations and determine the inflection points between liver enzymes and the risk of cognitive impairment.

However, it is imperative to recognize the potential limitations of this study. First, using a cross-sectional design precludes the establishment of causality, necessitating a cautious interpretation of the current findings. Second, while the extensive data collection conducted by NHANES enables the examination of established confounding variables, residual confounders cannot be entirely ruled out. Furthermore, it is important to note that our study lacked a clinical examination capable of diagnosing and further categorizing cognitive impairment. Finally, some studies have indicated that longitudinal alterations in liver enzymes may predict cognitive impairment [20,21]. Consequently, additional prospective investigations are required to explore the influence of both baseline liver enzyme levels and variability on the progression of cognitive impairment.

## 5 Conclusion

This study examined the independent relationship between liver enzymes and cognitive performance. Negative associations were observed between ALP, AST/ALT ratio, and GGT levels and cognitive performance. Conversely, a positive association was found between ALT levels and cognitive performance. Furthermore, this study identified non-linear relationships and their respective thresholds with cognitive performance. Both physical and nutritional interventions have the potential to exert a significant impact on liver enzymes and mitigate their

adverse effects on cognitive function [39,56]. Future research should incorporate interventional studies to evaluate the potential role of liver enzymes in cognitive decline.

## Supporting information

**S1 Table. The cutoff points of the CERAD test, AFT, and DSST adjusted based on age.**
(DOCX)

**S2 Table. Characteristics of liver enzymes and the CERAD test, AFT, and DSST among participants from NHANES 2011–2014 (N = 2764).**
(DOCX)

**S3 Table. The associations between liver enzymes and different dimensions of cognitive performance (CERAD Test, AFT and DSST) in Model 1 and Model 2 among participants from NHANES 2011–2014 (N = 2764).**
(DOCX)

**S4 Table. Subgroup analysis of the association between quartiles of ALP and cognitive performance.**
(DOCX)

**S5 Table. Subgroup analysis of the association between quartiles of ALT and cognitive performance.**
(DOCX)

**S6 Table. Subgroup analysis of the association between quartiles of AST/ALT ratio and cognitive performance.**
(DOCX)

**S7 Table. Subgroup analysis of the association between quartiles of GGT and cognitive performance.**
(DOCX)

**S8 Table. The associations between ALP and different dimensions of cognitive performance (N = 2747, sensitivity analysis).**
(DOCX)

**S9 Table. The associations between ALT and different dimensions of cognitive performance (N = 2746, sensitivity analysis).**
(DOCX)

**S10 Table. The associations between AST/ALT and different dimensions of cognitive performance (N = 2753, sensitivity analysis).**
(DOCX)

**S11 Table. The associations between GGT and different dimensions of cognitive performance (N = 2736, sensitivity analysis).**
(DOCX)

## Acknowledgments

We express our profound gratitude to all individuals who participated in this study and all those who contributed their efforts for this study.

## Author Contributions

**Conceptualization:** Yan-Li Zhang, Li-Ming Yang, Jun-Hong Guo.

**Formal analysis:** Bo Yang, Jie Miao, Chen Su, Zhi-Gang Cui.

**Methodology:** Bo Yang, Jun-Hong Guo.

**Supervision:** Li-Ming Yang, Jun-Hong Guo.

**Writing – original draft:** Yan-Li Zhang, Shi-Ying Jia.

**Writing – review & editing:** Li-Ming Yang, Jun-Hong Guo.

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
