## [Decision Letter · Decision Letter 0]

1 Apr 2024

PONE-D-24-02163Non-linear association of liver enzymes with cognitive performance in the elderly: A cross-sectional studyPLOS ONE

Dear Dr. Guo,

Thank you for submitting your manuscript to PLOS ONE. After careful consideration, we feel that it has merit but does not fully meet PLOS ONE’s publication criteria as it currently stands. Therefore, we invite you to submit a revised version of the manuscript that addresses the points raised during the review process. Please add more details on the statistical analysis performed by you.

We look forward to receiving your revised manuscript.

Kind regards,

Suyan Tian

Academic Editor

PLOS ONE

Journal Requirements:

2. Thank you for stating the following financial disclosure: " This work was supported by the Shanxi Province Basic Research Program (202303021212373)." 

Reviewers' comments:

Reviewer's Responses to Questions

**Comments to the Author**

1. Is the manuscript technically sound, and do the data support the conclusions?

Reviewer #1: Yes

2. Has the statistical analysis been performed appropriately and rigorously? 

Reviewer #1: N/A

3. Have the authors made all data underlying the findings in their manuscript fully available?

Reviewer #1: Yes

4. Is the manuscript presented in an intelligible fashion and written in standard English?

Reviewer #1: Yes

5. Review Comments to the Author

Reviewer #1: In this paper, the nonlinear relationship between liver enzymes and cognitive function is studied from a new Angle, the correlation and inflection point between the two are found, and the advantages and disadvantages between the two are better discussed. But there are still many problems in the article, I hope the author can modify it.

1. The BMI classification and covariates in Table 1 are inconsistent, please proofread the article and data analysis to see whether the classification is correct.

2. Table 1 is used as the baseline table. Please make an overall description of the total number of people and whether they are sick, and modify the counting data as NO.(%) for easy reference.

3. It can be seen from the flow chart that the author excluded participants with missing variables, but it is still reflected in Table 1 that some participants have multiple covariables with missing values. It is suggested to delete participants with missing variables and take a complete data set for statistical analysis.

4. Table 1 does not provide statistical description of P value, please add.

5. Table 2-5 does not conform to statistical logic. The left side is the independent variable and the right side is the dependent variable. Please revise it and combine Table 2-5 into one table. In this paper, the nonlinear relationship between liver enzymes and cognitive function is studied from a new Angle, the correlation and inflection point between the two are found, and the advantages and disadvantages between the two are better discussed. But there are still many problems in the article, I hope the author can modify it.

1. The BMI classification and covariates in Table 1 are inconsistent, please proofread the article and data analysis to see whether the classification is correct.

2. Table 1 is used as the baseline table. Please make an overall description of the total number of people and whether they are sick, and modify the counting data as NO.(%) for easy reference.

3. It can be seen from the flow chart that the author excluded participants with missing variables, but it is still reflected in Table 1 that some participants have multiple covariables with missing values. It is suggested to delete participants with missing variables and take a complete data set for statistical analysis.

4. Table 1 does not provide statistical description of P value, please add.

5. Table 2-5 does not conform to statistical logic. The left side is the independent variable and the right side is the dependent variable. Please revise it and combine Table 2-5 into one table.

6. Please supplement the statistical method of the data in the first column of Table 2-5 in the previous data analysis (whether the independent variable is a continuous variable or a categorical variable after the four categories) and which statistical method P for trend is the P value.

7. The author selected three models for data fitting, among which model 3 excluded the influence of confounding factors, reflecting the true correlation between liver enzymes and cognition in the presence of confounding factors. However, in most of the result descriptions, the author did not describe the contents with statistical differences in Model 3, and paid more attention to model 1. Moreover, the association between grouping variables between Q1-Q4 and cognition was not described, so the author was requested to modify the result description.

8. According to Table 2-5 and S3-6, there are statistical differences in some results Q2-Q4, but P for trend is not statistically significant. Is the analysis wrong? Please refer to a published article, use the correct statistical method and do a good analysis.

9. The discussion on logisitcs results mostly focused on the statistical results of model 1, but we mainly focused on the statistical significance of model 3 after adjusting covariates. Please adjust the relevant discussion.

10. The discussion of the same and different results of logistics and GAM can be added, so as to highlight the advantages of GAM model.

6. PLOS authors have the option to publish the peer review history of their article (what does this mean?). If published, this will include your full peer review and any attached files.

Reviewer #1: **Yes: **shan liu

---

## [Author Response · Author response to Decision Letter 0]

21 May 2024

Dear Editor and Reviewers:

We would like to extend our most sincere appreciation to the editor and the reviewers' valuable comments on our manuscript (PONE-D-24-02163, Non-linear association of liver enzymes with cognitive performance in the elderly: A cross-sectional study). According to the valuable comments, we have improved the quality of our work, and highlighted the changes with red text in the revised manuscript. Please see the uploaded word file “Response to reviewers” for a point-by-point response to the comments.

---

## [Decision Letter · Decision Letter 1]

4 Jun 2024

PONE-D-24-02163R1Non-linear association of liver enzymes with cognitive performance in the elderly: A cross-sectional studyPLOS ONE

Dear Dr. Guo,

Thank you for submitting your manuscript to PLOS ONE. After careful consideration, we feel that it has merit but does not fully meet PLOS ONE’s publication criteria as it currently stands. Therefore, we invite you to submit a revised version of the manuscript that addresses the points raised during the review process.

Please address the points raised by the reviewer, particularly the last one.

We look forward to receiving your revised manuscript.

Kind regards,

Suyan Tian

Academic Editor

PLOS ONE

Journal Requirements:

Reviewers' comments:

Reviewer's Responses to Questions

**Comments to the Author**

1. If the authors have adequately addressed your comments raised in a previous round of review and you feel that this manuscript is now acceptable for publication, you may indicate that here to bypass the “Comments to the Author” section, enter your conflict of interest statement in the “Confidential to Editor” section, and submit your "Accept" recommendation.

Reviewer #1: (No Response)

2. Is the manuscript technically sound, and do the data support the conclusions?

Reviewer #1: Yes

3. Has the statistical analysis been performed appropriately and rigorously? 

Reviewer #1: N/A

4. Have the authors made all data underlying the findings in their manuscript fully available?

Reviewer #1: Yes

5. Is the manuscript presented in an intelligible fashion and written in standard English?

Reviewer #1: Yes

6. Review Comments to the Author

Reviewer #1: I am very grateful to the author for adopting my previous suggestions to revise the paper in large quantities, and I also thank the author for his careful efforts to make this paper more perfect in many details. However, after this detailed reading, I have some suggestions that I hope the author can adopt.

1. There are unrecorded descriptions in Table 1, such as BMI, smoking, etc., but the descriptions of covariables in 2.4 Covariates are not explained, please add. Are data missing from these variables included in logisitcs and GAM? How many people are included in logisitcs and GAM?

2.3.2.1 Description of the Relationship between ALP and cognitive performance. Please describe the relationship in the order listed in the table.

3. Table 3 suggests showing the results of Model 3 directly, and the results of model 12 can be attached.

4. As far as I know, different dimensions of cognitive performance include CERAD-IR, Consortium to Establish a Registry for Alzheimer's Disease Immediate Recall, CERAD-DR, Consortium to Establish a Registry for Alzheimer's Disease Delayed Recall, AFT, animal fluency test; DSST, digit symbol substitution test.Why are there only three dimensions in this article?

7. PLOS authors have the option to publish the peer review history of their article (what does this mean?). If published, this will include your full peer review and any attached files.

Reviewer #1: **Yes: **shan liu

---

## [Author Response · Author response to Decision Letter 1]

14 Jun 2024

Please see the uploaded attachment" Response to Reviewers", thanks!

---

## [Decision Letter · Decision Letter 2]

25 Jun 2024

Non-linear association of liver enzymes with cognitive performance in the elderly: A cross-sectional study

PONE-D-24-02163R2

Dear Dr. Guo,

We’re pleased to inform you that your manuscript has been judged scientifically suitable for publication and will be formally accepted for publication once it meets all outstanding technical requirements.

Kind regards,

Suyan Tian

Academic Editor

PLOS ONE

Additional Editor Comments (optional):

Reviewers' comments:

Reviewer's Responses to Questions

**Comments to the Author**

1. If the authors have adequately addressed your comments raised in a previous round of review and you feel that this manuscript is now acceptable for publication, you may indicate that here to bypass the “Comments to the Author” section, enter your conflict of interest statement in the “Confidential to Editor” section, and submit your "Accept" recommendation.

Reviewer #1: All comments have been addressed

2. Is the manuscript technically sound, and do the data support the conclusions?

Reviewer #1: Yes

3. Has the statistical analysis been performed appropriately and rigorously? 

Reviewer #1: Yes

4. Have the authors made all data underlying the findings in their manuscript fully available?

Reviewer #1: Yes

5. Is the manuscript presented in an intelligible fashion and written in standard English?

Reviewer #1: Yes

6. Review Comments to the Author

Reviewer #1: (No Response)

7. PLOS authors have the option to publish the peer review history of their article (what does this mean?). If published, this will include your full peer review and any attached files.

Reviewer #1: **Yes: **shan liu

---

## [Editor Report · Acceptance letter]

15 Jul 2024

PONE-D-24-02163R2 

PLOS ONE

Dear Dr. Guo, 

I'm pleased to inform you that your manuscript has been deemed suitable for publication in PLOS ONE. Congratulations! Your manuscript is now being handed over to our production team.

Kind regards, 

on behalf of

Dr. Suyan Tian 

Academic Editor

PLOS ONE